# Ain't Nobody Got Time for Coding: Structure-Aware Program Synthesis from Natural Language

## Abstract

Program synthesis from natural language (NL) is practical for humans and, once technically feasible, would significantly facilitate software development and revolutionize end-user programming. We present SAPS, an end-to-end neural network capable of mapping relatively complex, multi-sentence NL specifications to snippets of executable code. The proposed architecture relies exclusively on neural components, and is trained on abstract syntax trees, combined with a pretrained word embedding and a bi-directional multi-layer LSTM for processing of word sequences. The decoder features a doubly-recurrent LSTM, for which we propose novel signal propagation schemes and soft attention mechanism. When applied to a large dataset of problems proposed in a previous study, SAPS performs on par with or better than the method proposed there, producing correct programs in over 92% of cases. In contrast to other methods, it does not require post-processing of the resulting programs, and uses a fixed-dimensional latent representation as the only interface between the NL analyzer and the source code generator.

**Keywords:** Program synthesis, tree2tree autoencoders, soft attention, doubly-recurrent neural networks, LSTM.

## 1 Introduction

Program synthesis consists in automatic or semi-automatic (e.g., interactive) generation of programs from specifications, and as such can be posed in several ways. It is most common to assume that specification has the form of input-output pairs (tests), resembling learning from examples, which is by definition limited by its inductive nature: even if a program passes all provided tests, little can be said about generalization for other inputs. This is one of the rationales for synthesis from formal specifications, which are typically expressed as a pair of logical clauses (a *contract*), a precondition and a postcondition. Programs so synthesized are correct by construction, but the task is NP-hard, so producing programs longer than a few dozen of instructions becomes computationally challenging. Synthesis from formal specifications is also difficult for programmers, because writing a correct and complete specification of a nontrivial program can be actually *harder* than its implementation.

From the practical perspective, the most intuitive and convenient way of specifying programs is natural language (NL). This way of formulating synthesis tasks has been rarely studied in the past, but the recent progress in NLP and deep neural networks made it more realistic, as confirmed by a few studies we review in Section 5. Here, we propose Structure-Aware Program Synthesis (SAPS[1]), an end-to-end approach to program synthesis from natural language. SAPS receives a short NL description of requested functionality and produces a snippet of code in response. To that aim, we combine generic word and sequence embeddings with doubly-recurrent decoders trained on abstract syntax trees (ASTs), with the following original contributions: (i) folding the entire NL specification into a single point in a fixed-dimensional latent space, which is then mapped by decoder onto the AST of a code snippet; (ii) modular architecture that facilitates usage of pretrained components, which we illustrate by pretraining the decoder witihn an autoencoder architecture; (iii) new signal propagation strategies in the decoder; and (iv) a 'soft attention' mechanism over the points in latent space. Last but not least, the implementation engages a form of dynamic batching for efficient processing of tree structures.

---

[1]Pun intended.

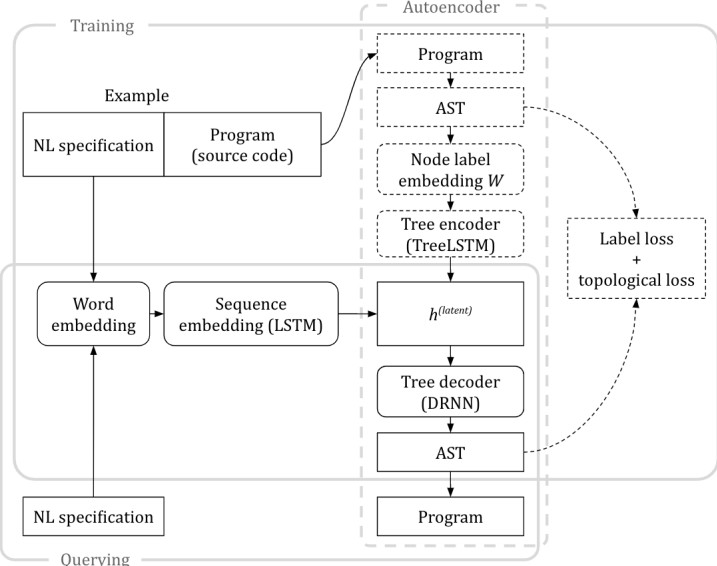

Figure 1: Overall architecture of SAPS. Rectangles mark data entities, rounded boxes are operations. The dashed components are used only in the variants of the method that involve autoassociative pretraining of the decoder.

## 2 SAPS ARCHITECTURE

Once trained, SAPS accepts a short NL specification as input and produces in response an AST of a snippet of code (program). It comprises of three main components (Fig. 1): (i) a word embedding for preprocessing of the specification, (ii) mapping of the embedded word sequence tokens to a fixed-size latent representation $h^{(latent)}$, and (iii) a decoder that unfolds $h^{(latent)}$ into an AST. Depending on the variant of the methods, the decoder may be trained from scratch or be pretrained within a tree2tree autoencoder (the dashed blocks in the figure).

### 2.1 WORD EMBEDDING

The NL specifications proposed in Polosukhin & Skidanov (2018), which we use in the experimental part, include only standard English vocabulary. Given that, we rely on the smallest pretrained GloVe embedding that covers all terms occurring in the considered dataset, more specifically the Common Crawl, which has been trained on generic NL database comprising 42B tokens by Pennington et al. (2014), has vocabulary size of 1.9M tokens, and embeds the words in a 300-dimensional space. Given an input NL query phrase of $n$ tokens, this module produces a sequence of $n$ 300-dimensional vectors $(q_1, \ldots, q_n)$. In order to be able to train the network with mini-batches of data, we pad each sequence in the mini-batch with a special out-of-vocabulary value.

### 2.2 SEQUENCE EMBEDDING

To map the sequences of word embeddings of the NL specification (300-dimensional vectors $(q_1, \ldots, q_n)$) to the latent space $h^{(latent)}$, we employ a multilayer bidirectional LSTM (Schuster & Paliwal, 1997; Graves et al., 2005). Because some variants of SAPS engage pretraining, the concateneted outputs of the both vertical and horizontal LSTMs is passed through a $\mathtt{tanh}$ layer in order to match the $h^{(latent)}$ dimensionality and the $(-1, 1)$ range of values produced by the tree encoder.

Internally, our sequence embedding is a 4-layer LSTM: there are four forward cells stacked upon each other, i.e. each consecutive cell receives the output of the previous cell as input, and analogously a stack of four backward cells. The final state of the topmost forward cell (reached after the forward pass over the input sequence) and the final state of the topmost backward cell (reached after the backward pass over the input sequence) are concatenated to form the final output.

## 2.3 TREE DECODER

As signaled earlier, the decoder can be trained from scratch or pretrained within an autoencoder (dashed part of Fig. 1), and its internal architecture is the same in both scenarios. We proceed with presenting it as a part of the autoenceoder, as we find it more convenient to the reader. Our tree2tree autoencoder architecture, which we originally introduced in (Anonymous, 1999), is based on TreeLSTM Tai et al. (2015) for encoding and Doubly-Recurrent NN (DRNN) Alvarez-Melis & Jaakkola (2017) for decoding. However it diverges from the latter work in (i) using only the latent vector for passing the information between encoder and decoder, and (ii) in new ways in which decoder's state is being maintained when generating trees.

**Encoder**. A single training example is $(x, x) \in X_{AST} \times X_{AST}$, where $x$ is an AST of a snippet of code. Given $(x, x)$, the encoder starts by encoding the label of every node $x_j$ in $x$ using one-hot encoding $v_j = V(x_j)$; for the experiments presented in Section 4, there are 72 node labels, so $|v_j| = 72$. Then, following Mikolov et al. (2013), we map $v_j$s, for each node in $x$ independently, to a lower-dimensional space, using a learnable embedding matrix $W$: $y_j = W v_j$. The result is a tree $y$ of node label embeddings $y_j$, isomorphic to $x$.

The encoder network folds the tree bottom-up, starting from the tree leaves, aggregating the hidden states $h_j$, and ending up at the root node. Contrary to some other use cases, where node arity (the number of children) is the same for all internal nodes within a tree, in ASTs nodes can have arbitrary arity. For this reason, we apply the recurrent TreeLSTM Tai et al. (2015) that can merge the information from any number of children.

The hidden states and cell states are initialized with zeroes. For each node, we first compute the sum of the hidden states of its children $C(j)$:

$$\widetilde{h_j} = \sum_{k \in C(j)} h_k \tag{1}$$

For leaf nodes, $\widetilde{h_j} = 0$. Then we compute the values of input, forget and output gates:

$$i_j = \sigma(W_i y_j + U_i \widetilde{h_j} + b_i) \tag{2}$$

$$f_{jk} = \sigma(W_f y_j + U_f h_k + b_f) \tag{3}$$

$$o_j = \sigma(W_o y_j + U_o \widetilde{h_j} + b_o) \tag{4}$$

$$u_j = \tanh(W_u y_j + U_u \widetilde{h_j} + b_u) \tag{5}$$

where $\sigma$ is the sigmoid function, and $W_i, W_f, W_o$ and $W_u$ and $b_i, b_f, b_o$ and $b_u$ are, respectively, learnable weights and biases. Note that the values of $i_j, o_j$ and $u_j$ depend on the aggregated children's states ($\widetilde{h_j}$), whereas $f_{jk}$ is computed child-wise, yielding a separate output $f_{jk}$ of forgetting gate for each child of the current node.

We then update the state $c_j$ of the memory cell of the current node:

$$c_j = i_j \odot u_j + \sum_{k \in C(j)} f_{jk} \odot c_k \tag{6}$$

where $\odot$ stands for elementwise multiplication, and where we assume that the initial states of $c_j$s of non-existent children of leaves are zero. Next, we compute the hidden state of the current node:

$$h_j = o_j \odot \tanh c_j, \tag{7}$$

which is then recursively aggregated by the above procedure. Once this process reaches the root node, the obtained state $h_j$ becomes the latent vector, i.e. $h^{(latent)}(x)$. Additionally, we have applied *layer normalization* (Ba et al., 2016) in the same manner as to traditional LSTM.

**Decoder**. Our decoder is a doubly recurrent neural network (DRNN) proposed in Alvarez-Melis & Jaakkola (2017). It comprises two LSTM cells, which separately capture the *vertical* and *horizontal*

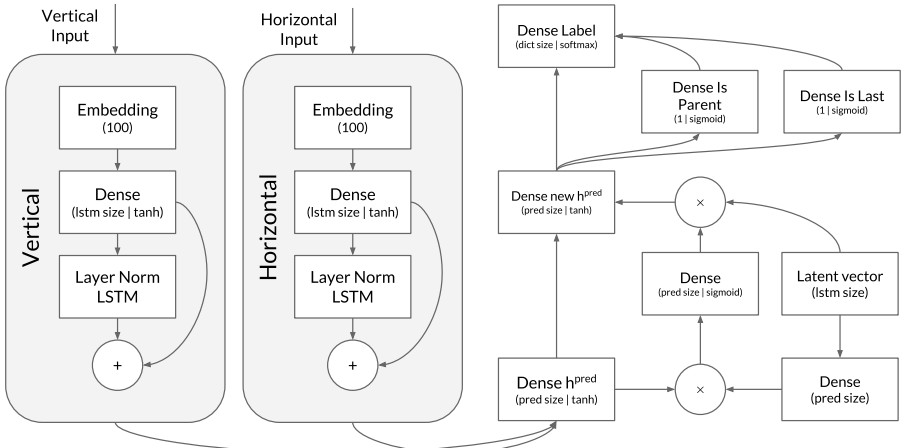

Figure 2: The modified DRNN cell consists of two LSTM blocks, each comprising embedding, input projection layer to adapt the input size to LSTM size, layer normalization LSTM cell, and residual skip connection between the projection and the output layers. DRNN cell uses the latent vector to enrich the prediction hidden state by the information that might have been lost during hierarchical processing.

traversal of the resulting AST tree. The state of the former $(h_i^a)$ is passed vertically downwards from parents to children, and of the latter $(h_i^f)$ horizontally left-to-right to consecutive siblings.

Importantly, we propose to augment the DRNN with modified schemes of state propagation and novel attention mechanism applied to the latent vector (see Fig. 2). In the original DRNN, the state of the horizontal LSTM was reset before processing the children of each node. However, preliminary experiments with Python ASTs we conducted in (Anonymous, 1999) have shown that discarding this resetting step, i.e. allowing the state to propagate between children of different parent nodes, may bring significant improvements. Our working explanation for this observation is that ASTs capture only program *syntax*, while pieces of code can be *semantically* related even if they do not reside in a common part of an AST. By not resetting the state of the horizontal LSTM, we provide it with more context that would be otherwise unavailable.

This observation inclined us to consider three variants of DRNN, denoted in the following as (Fig. 3): (i) *H*, in which only the state of the horizontal LSTM is propagated (alongside the entire depth

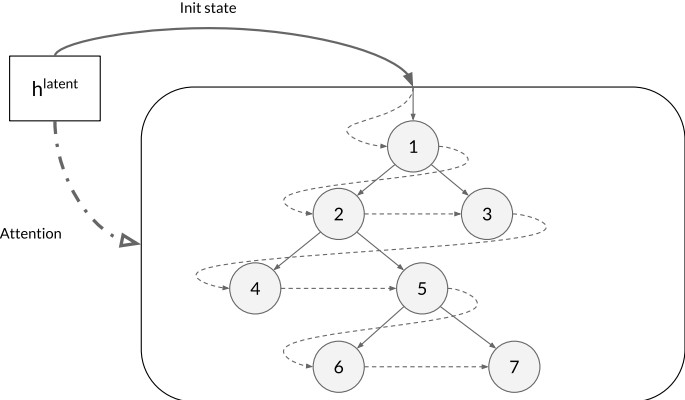

Figure 3: Propagation of the hidden states and the latent vector during decoding. The dashed line illustrates the traversal of the horizontal state, the solid line propagation of the vertical state. The latent vector is used to initialize both states, but has also impact on subsequent decoding steps via an attention mechanism.

level of a tree), (ii) *V*, in which only the state of the vertical LSTM is propagated, (iii) *VH*, in which states of both LSTMs are propagated.

These three DRNN variants vary only in the above aspect of propagation (or, conversely, state re-setting), while all remaining processing stages remain the same. We start by initializing both states of the vertical and the horizontal LSTM cells $h_i^a, h_i^f$ with $h^{(latent)}(x)$. Alvarez-Melis & Jaakkola (2017) suggest to initialize only vertical hidden state with latent vector, however, our modification concerning the transfer of horizontal states in breadth-first order implies the same initialization approach also to the horizontal state. The DRNN merges both states of LSTMs in a *prediction state*:

$$h_i^{(pred)} = \tanh(U^f h_i^f + U^a h_i^a), \tag{8}$$

where $U^f$ and $U^a$ are learnable parameters. $h_i^{(pred)}$ is then mapped to the probability distribution of node's label

$$l_i = \text{softmax}(W h_i^{(pred)} + \alpha_i v^a + \varphi_i v^f), \tag{9}$$

where $|l_i| = |v_i|$ (see the description of the encoder), and $v^a$ and $v^f$ are learnable parameters applied to *is-parent* flag ($\alpha_i$) and *is-last-child* flag ($\varphi_i$). In training, these are used for *teacher forcing*, i.e. replaced with ground truth data according to the shape of the input tree $x$; in inference mode, the flags are calculated as:

$$\alpha_i^{(pred)} = \sigma(u^a h_i^{(pred)}) \tag{10}$$

$$\varphi_i^{(pred)} = \sigma(u^f h_i^{(pred)}) \tag{11}$$

where $u^a$ and $u^f$ are learnable parameters.

In training, we minimize the sum of *label loss* and *topological loss*:

$$L(x) = \sum_i L^{xent}(l_i, V(x_i)) + L^{bxent}(\alpha_i^{(pred)}, \alpha_i) + L^{bxent}(\varphi_i^{(pred)}, \varphi_i) \tag{12}$$

where $L^{xent}(l_i, V(x_i))$ is the categorical cross-entropy loss between the ground truth label $V(x_i)$ and the label $l_i$ produced by the DRNN, and $L^{bxent}$ is the binary cross-entropy, applied separately to the *is-parent* flag and *is-last-child* flag. *Teacher forcing* makes the result tree structure always equal to the target one allowing us to calculate the loss function for each node.

**Soft attention mechanism**. Because successive iterations of the DRNN may arbitrarily modify its prediction state $h_i^{(pred)}$, the information from the encoder may fade away with time, making it harder to reproduce the input tree $x$. A range of works, most relevantly Chen et al. (2018), have shown that this loss of information can be supplemented with an *attention mechanism*. However, the specific attention architecture proposed there assumes that the window of attention scans the corresponding nodes of the input tree $x$, which implies that $x$ has to be available also during decoding. We cannot assume this in our setting, where the decoder is intended to be ultimately detached from the encoder and receive only the latent vector $h^{latent}$ as input (Fig. 1). Therefore, we come up with an alternative attention mechanism that relies only on $h^{latent}$. We define a *soft attention window* as

$$A = \sigma((h^{(latent)} U^{(A)} \odot h_i^{(pred)}) W^{(A)}) \tag{13}$$

and then use it to update the prediction state as follows (compare to the attention-less update formula (8)):

$$h_i^{(pred)} = \tanh((A \odot h^{(latent)}) C^{(A)} + h_i^{(pred)} C^{(pred)}) \tag{14}$$

where $U^{(A)}$, $W^{(A)}$, $C^{(A)}$ and $C^{(pred)}$ are learnable parameters. We expect this mechanism to learn focusing on the parts of $h^{(latent)}$ that carry the information that is relevant for reproduction of tree structure, particularly at the deeper levels. Let us emphasize that this mechanism *scales* the original tensor, rather than explicitly *selecting* some of its elements, and as such can be likened to a *gating function*; nevertheless, we refer to it as 'soft attention' by analogy to certain models used in visual information processing.

```
program       ::= symbol
symbol        ::= constant | argument | function_call | function | lambda
constant      ::= number | string | True | False
function_call ::= (function_name arguments)
function      ::= function_name
arguments     ::= symbol | arguments , symbol
function_name ::= Reduce | Filter | Map | Head | + | - | ...
lambda        ::= lambda function_call
```

Figure 4: The grammar of the AlgoLisp DSL from (Polosukhin & Skidanov, 2018).

## 3 PROGRAM SYNTHESIS TASKS

We assess the performance of SAPS on the set of NL-based program synthesis tasks proposed in (Polosukhin & Skidanov, 2018). The domain-specific language (DSL) used in this suite is Lisp-based AlgoLisp, defined by the grammar in Fig. 4. A single program is composed as a nested list of instructions and operands of three types: `string`, `Boolean` and `function`. The language features a number of standard functions (cf. the last-but-one production in the grammar). The motivation for choosing this dataset was the large number of examples and brevity of NL specifications (in contrast to the programming contest data we used in (Anonymous, 1999)). Moreover, the simplicity of syntax causes AlgoLisp programs to be almost equivalent to their AST trees (in contrast to, e.g., Python we used in previous studies, where even a minimal program may translate into an AST of a dozen or two nodes).

The dataset comprises 99506 examples, each being a pair of NL specification and the corresponding AlgoLisp program. To handle variables, like $a$ in the example shown in Table 1, we use the same approach as Polosukhin & Skidanov (2018), i.e., each occurrence of a new variable allocates a new *placeholder*, and the placeholders extent the vocabulary of tokens used in the NL embedding. Placeholders are common for all examples, i.e., the first occurrences of variables in all programs are mapped to the same, first placeholder.

For conformance with the cited work, we split the data into training set of 79214 examples, validation set (9352 examples) and test set (10940 examples). The average lengths of NL specification and associated code are, respectively, 38 and 24 tokens. The shortest and longest NL description comprise 8 and 164 tokens respectively (2 and 217 for code snippets).

It may be worth noting that, in contrast to the relatively large number of examples available, the vocabularies of terms are rather humble: there are only 281 unique terms/tokens in the NL specifications, and only 72 unique terms in the AlgoLisp programs. The NL vocabulary used in specifications is strongly related to programming, containing words rarely used in everyday language, lie *prime, inclusive, incremented* etc. Nevertheless, they are all present in the Common Crawl GloVe embedding described in Section (Sect. 2.1).

**Dataset inconsistency**. After closer examination, we identified inconsistency in the dataset: for a about 10% of tasks (in the test set), some tests are not passed even by the ground truth program (we verified that using the AlgoLisp interpreted published in (Polosukhin & Skidanov, 2018)). To address this issue, we decided to use two performance measures: one based on the unmodified test and validation sets, in order to maintain comparability to results presented in that study, and another one using a filtered test set, containing only the tasks that were free from the above problem. Note however that this issue does not affect the training of SAPS, as it does not involve tests.

Table 1: An example AlgoLisp program (bottom) and its NL specification (left).

| |
|---|
| *You are given an array $a$. Find the smallest element in $a$, which is strictly greater than the minimum element in $a$* |
| `(reduce (filter a (partial0 (reduce a inf) <)) inf min)` |

Table 2: Percentage of perfectly synthesized programs (i.e., syntactically identical to the target ones) for SAPS configurations **trained from scratch** and latent vector of length 256. Total number of trained parameters, in millions, used in each configuration is shown in the most right column.

| Model | Validation | Test | Number of parameters [M] |
|---|---|---|---|
| SAPS V | 0% | 0% | 5.06 |
| SAPS V Att | 0% | 0% | 5.17 |
| SAPS H | 84.31% | 78.92% | 5.06 |
| SAPS H Att | 89.54% | 86.20% | 5.17 |
| SAPS VH | 93.65% | 89.10% | 5.62 |
| SAPS VH Att | **93.73%** | **92.36%** | 5.73 |

## 4 EXPERIMENT

We examine a range of SAPS configurations that vary along the dimensions mentioned earlier: (i) decoder state propagation schemes: vertical (*V*), horizontal (*H*), and both (*VH*), (ii) presence/absence of the soft attention mechanism (*Att*), (iii) pretraining the decoder (*pre*) or training it from scratch (*pre*). Atop of that, we consider also a range of sizes of $h^{(latent)}$ (64, 128, and 256). We train all components using Adam optimizer (Kingma & Ba, 2014) with initial learning rate set to $5 \times 10^{-4}$ and then reduced exponentially at rate 0.975 per every two epochs. To lessen the risk of overfitting, we applied $L_2$ regularization to all trainable parameters, except for biases and parameters from layer normalization.

**Training SAPS from scratch**. In Table 2, we present the predictive accuracy of synthesis of SAPS trained end-to-end from scratch in six configurations that vary on the first two of above-mentioned dimensions, i.e. propagation scheme and presence of attention mechanisms. Model's output is considered correct if the produced AST tree represents exactly the syntax of the target program. All these models use the latent vector of size 256. The synergy between the vertical and horizontal propagation schemes is quite evident: using them simultaneously boosts the accuracy by a few percent points compared to the configurations where just one of them is engaged. The effect of the attention mechanism is also overwhelmingly positive, particularly on the test set. Absence of propagation of the horizontal state substantially cripples SAPS, which was expected due to the fact that from the same vertically propagated state the network tries to produce different nodes.

**Pretraining of decoder**. In the second experiment, we ask whether the best SAPS configuration identified above (SAPS VH Att with latent representation of size 256) can benefit from pretraining of the DRNN decoder in the autoencoder architecture (Fig. 1). To this aim, we first train the autoencoder to reproduce AST trees on all programs from the training set until saturation, then combine the trained DRNN decoder obtained in this way with the remaining modules to form the SAPS pipeline, which is then trained end-to-end on the (NL specification, program) pairs. Comparing the first row of Table 3 with Table 2 suggests that pretraining may be indeed helpful: although the pretrained model attains marginally worse accuracy on the test set, it beats the trained from scratch on the validation set by almost one percent point. The answer to the above question is not therefore definitive, nevertheless we consider this result as a strong case in favor of pretraining.

We examined also the capabilities of pretrained models with smaller latent size (128 and 64). Their performances (Table 3) were significantly worse, which indicates that the latent size in the first experiment was set adequately. Though this may suggests that further improvements could be achieved by increasing the latent size beyond 256, we keep this setting for the rest of this study for the sake of comparability.

In the above configurations, the decoder (taken from the trained autoencoder) learns alongside with the sequence embedding LSTM. For comparison, we consider also a variant dubbed SAPSpre VH Att (256) Fixed, where the latent vectors from the pretrained autoencoder model serve as (fixed) target (desired output) for the sequence embedding LSTM. Crucially then, in this configuration the pretrained decoder remains fixed during the latter phase of training. As expected, the obtained accuracy is inferior to other configurations, which clearly indicates that mutual co-adaptation of sequence embedding and pretrained decoder is essential. Nevertheless, SAPSpre VH Att (256) Fixed still outperforms the approach based only on input-output pairs from (Polosukhin & Skidanov,

Table 3: Percentage of perfectly synthesized programs (i.e., syntactically identical to the target ones) for **pretrained** SAPS configurations. Size of the latent vector given in parentheses. The rightmost two columns show the percentage of programs perfectly reproduced by the corresponding autoencoder (AC) used for pretraining.

| Model | Validation | Test | AC-Validation | AC-Test |
|---|---|---|---|---|
| SAPSpre VH Att (256) | **94.59%** | **92.14%** | 72.98% | 68.42% |
| SAPSpre VH Att (128) | 82.59% | 77.07% | 67.06% | 60.18% |
| SAPSpre VH Att (64) | 58.65% | 52.22% | 50.24% | 39.36% |
| SAPSpre VH Att (256) Fixed | 14.91% | 14.39% | | |

Table 4: Percentage of programs passing all tests. Results for methods other than SAPS cited from Polosukhin & Skidanov (2018).

| Model | Validation | Test |
|---|---|---|
| Attentional Seq2Seq | 54.4% | 54.1% |
| Seq2Tree | 61.2% | 61.0% |
| SAPSpre VH Att (256) | **86.67%** | **83.80%** |
| Seq2Tree + Search | 86.1% | 85.8% |

Table 5: Percentage of programs passing all tests, using all examples provided in the AlgoLisp database (*All tests*), and using only the tests that are consistent with the target program (*Only passable tests*). See the main text for detailed explanation.

| Model | All tests | | | | Only passable tests | | | |
|---|---|---|---|---|---|---|---|---|
| | Validation | | Test | | Validation | | Test | |
| | Accuracy | 50-Accuracy | Accuracy | 50-Accuracy | Accuracy | 50-Accuracy | Accuracy | 50-Accuracy |
| SAPSpre VH Att (256) | 86.67% | 89.64% | 83.80% | 87.45% | 95.02% | 95.81% | 92.98% | 94.15% |
| SAPSpre VH Att (128) | 76.62% | 80.92% | 70.91% | 75.89% | 83.90% | 86.41% | 78.56% | 81.54% |
| SAPSpre VH Att (64) | 55.08% | 60.35% | 49.80% | 56.31% | 60.19% | 64.31% | 55.17% | 60.56% |

2018), which achieved 13.3% of accuracy in the best case, despite relying on an external search mechanism.

**Behavior of synthesized programs on input-output tests**. In Table 4 we juxtapose our best configuration, SAPSpre VH Att (256), with the results reported by Polosukhin & Skidanov (2018). This time the comparison metric is the percentage of programs that pass all tests provided (alongside with NL specifications and target programs) in the AlgoLips database prepared by the authors of that study, using their implementation of evaluation metric. Due to the inconsistencies in of tests in about 10% of examples in that dataset (Section 3), the numbers for our best SAPS configuration are smaller than those in Table 3, which normally would not be possible. Nevertheless, the comparison is fair, as all methods are tested on the same (faulty) tests and thus can be expected to be affected to the same extent. SAPS clearly outperforms the models based on Seq2Seq and Seq2Tree architectures. More importantly however, it performs on par with the Seq2Tree combined with external search algorithm, which was the best approach reported in the cited work (last row of the table). Though SAPS does not match its test-set performance, i.e. 85.8%, it is worse only by 2 percent point (and slightly better on the validation set), while being trained end-to-end using gradient, rather than with the help of a sophisticated search mechanism and input-output tests (recall that SAPS does not use tests at all).

To address the above-mentioned issue of dataset inconsistency (Section 3), we prepare a modified version of the AlgoList database, where all examples containing faulty, non-passable tests have been removed. In Table 5 we compare the results of the best configurations of SAPS applied to this dataset (right columns) to their performance on the original, unmodified AlgoLisp (left columns). For better insight, we report not only the percentage of programs that pass all tests (*Accuracy*), but also an analogous number for programs that pass at least 50% of tests (*50-Accuracy*). Expectedly, networks obtained higher scores when evaluated only on the tests that are free from the abovementioned issue,

and those scores are also higher than the percentages of programs that were syntactically identical to the target (Table 3). The latter fact should not be surprising, as a syntactically correct program has to pass all tests.

It may be worth mentioning that none of the considered trained configurations of SAPS produced any syntactically incorrect programs.

**Illustration of SAPS response to modifications of specification**. In Table 6 we present programs synthesized by SAPS for exemplary NL specifications and its response to various input modifications. The first part of the table illustrates how the generated source code changes as the specification is being simplified. Remarkably, the network produces correct output even when the input is very simplified and laconic. We evaluated also generalization by replacing operations (*multiplied → minimum*), applying different ranges of arrays, and complex modifications affecting multiple parts of NL specification simultaneously.

## 5 RELATED WORK

With the recent advent of powerful hardware and new methodological developments, the problem of program synthesis received increasing attention. Apart from the fundamental research directions mentioned in the Introduction, a few main approaches can be identified.

**Program synthesis from examples**. In addition to a large body of literature on heuristic synthesis of programs from examples (usually with genetic programming, a variant of evolutionary algorithm), recently a handful of analogous neural methods were introduced. Balog et al. (2016) used a neural network as a guide for a search algorithm, in order to predict the most probable code tokens, which greatly reduced the number of solutions that needed to be evaluated during search. Krawiec et al. (2017) used genetic programming to find a set of programs that match the formal specification, which ensures the correctness of generated programs for all possible inputs from specification. The

Table 6: The effects of modifications of NL specification. The first specification in each group is an original task from the validation set, and those that follow are its modified variants (modifications marked in bold). Row 1: Dropping words from the specification until the output program is not correct anymore. Row 2: Modifying a single keyword in the output program. Row 3: Change of entire subexpression in the output program. Row 4: multiple co-occurring modifications of specification. Except for the specification *a b, a + b*, all synthesized programs are consistent with specification.

| Specification | Synthesized program |
|---|---|
| *you are given numbers a and b, your task is to find a + b* | `(+, a, b)` |
| *you given numbers a b, your is find a + b* | `(+, a, b)` |
| *given **a numbers b**, find a + b* | `(+, a, b)` |
| *given a numbers b, a + b* | `(+, a, b)` |
| *a b, a + b* | **`(+,`** `(+, a, b),` **`c)`** |
| *you are given numbers a and b, your task is to find **a multiplied by b*** | `(*, a, b)` |
| *you are given numbers a and b, your task is to find **minimum a and b*** | `(**min**, a, b)` |
| *given a number a and an array of numbers b, find the length of the longest subsequence of range from **0 to a** inclusive that is a prefix of b* | `(reduce, (range, 0, (+, a, 1)), 0, (lambda2, (if, (==, arg2, (if, (<, arg1, (len, b)), (deref, b, arg1), 0)), (+, arg1, 1), arg1)))` |
| *given a number a and an array of numbers b, find the length of the longest subsequence of range from **1 to a** exclusive that is a prefix of b* | `(reduce, (range, 1, a), 0, (lambda2, (if, (==, arg2, (if, (<, arg1, (len, b)), (deref, b, arg1), 0)), (+, arg1, 1), arg1)))` |
| *given an array of numbers a, find median of values in a after only keeping first half* | `(deref, (sort, (slice, a, 0, (/, (len, a), 2))), (/, (len, (slice, a, 0, (/, (len, a), 2))), 2))` |
| *given an array of numbers a, find mean of values in a after only keeping second half* | `(/, (reduce, (slice, a, (/, (len, a), 2), (len, a)), 0, +), (len, (slice, a, (/, (len, a), 2), (len, a))))` |

method supplied itself with examples by collecting counterexamples resulting from iterative formal verification process. Devlin et al. (2017) create programs using noisy input-output pairs.

**Differentiable compilers**. Another approach is to, rather than synthesizing human readable code, embed the whole process implicitly in the neural substrate. These methods often utilize neural networks and could be seen as differentiable compilers. Graves et al. (2014) implement a Turing machine using neural network, endowing it with differentiable memory. A similar approach was proposed by Sukhbaatar et al. (2015), where a memory was also used.

**Synthesis from NL**. With the recent advances in NL understanding, a number of approaches utilizing NL descriptions for code synthesis were introduced. Rabinovich et al. (2017) and Yin & Neubig (2017) generate ASTs rather than plain source code, using recursive neural decoders specialized for AST generation. An interesting approach is presented by Ling et al. (2016), where the authors use a set of neural modules to generate source code from multimodal inputs (text and quantitative features) and apply their framework to a trading card game. Another related topic is *semantic parsing* (see, e.g., (**?**)), which puts particular emphasis on synthesis from NL and other informal specifications.

Last but not least, the main reference approach for this work is the neuro-guided approach proposed by Polosukhin & Skidanov (2018). The authors used a neural network to guide a Tree-Beam search algorithm and so prioritize the conducted search process. In the approach proposed there, a sequential encoder folds a short description in natural language, followed by a tree decoder that generates a set of most possible nodes in the AST. For each node, the proposed labels are evaluated by the search algorithm on a set of input-output tests. The process is repeated recursively for consecutive nodes of the AST tree. The authors used an attention mechanism (Bahdanau et al., 2014) on the entire input sequence – contrary to our approach, where attention concerns only the latent vector.

## 6 DISCUSSION

SAPS manages to achieve state-of-the-art test-set accuracy, on par with that of Polosukhin & Skidanov (2018), and does so with a bare neural model, without any search, additional postprocessing or other forms of guidance. This remains in stark contrast to that study, where network was queried repeatedly in a Tree-Beam search heuristics to produce the target program step by step, testing the candidate programs on provided tests (see Fig. 2 in that work). There are also many differences with respect to (Balog et al., 2016), where a network was used to prioritize search conducted by an external algorithm. SAPS's architecture can provide a similar level of quality with purely neural mechanisms. This seems conceptually interesting in itself; in particular, it is worth realizing that the fixed-dimensionality latent layer $h^{(latent)}$ implements a complete embedding for a large number of programs – those present in our training, validation and testing set, but possibly also others.

The fact that none of SAPS configurations ever produced a syntactically incorrect program (including the testing phase) suggests that SAPS captured well the *syntax* of AutoLisp and is likely to produce well-formed programs for other NL specifications – which was also corroborated by the effects of manipulations shown in Table 6. Moreover, it is also likely that SAPS has rather good insight into program *semantics*, which can be concluded from its behavior on the passable tests, summarized in the right part of Table 5. Even when a synthesized program is not identical to the target one, there is still a chance for it to be correct. For instance, for the test set, SAPSpre VH Att (256) produces a perfect copy of the target program in 92.14% of test examples (Table 3), while in total 92.98% all programs it synthesizes pass all passable tests (Table 5). Therefore, 0.84% of programs pass all tests despite being syntactically different from the target program (which translates into roughly one in 9 of such programs). It seems likely that in those cases the network came up with an alternative implementation of the target concept expressed by the NL specification. One should keep in mind, however, that passing the tests does not prove semantic equivalence. This could be achieved with formal verification, which we leave out for future work.

In relation to that, let us also note that a substantial fraction of programs that do not pass all tests, pass at least 50% of them (see the 50-Accuracy in Table 5). Since the a priori probability of passing a test by a program is miniscule, this suggests that even the imperfect programs feature useful code snippets that make them respond correctly to some tests.

On the other hand, it would be naive to expect SAPS to scale well for much longer NL specifications and/or target programs (the median length of the former was 36 words, and of the latter

20 tokens). That would arguably require augmenting the method with additional modules, for instance an attention mechanism for interpretation of the NL specification, as used by Polosukhin & Skidanov (2018). Such functionality seems particularly desirable for tasks that involve complex, multi-sentence NL specifications. Note however that any additional mechanisms that circumvents the latent layer makes this architecture less modular, which may be undesirable, particularly for pretraining. For instance, our autoencoder can be now potentially pretrained on a (possibly large) collection of programs without NL specifications. Introducing direct attention from the DRNN to the NL layers would break that independence.

We attribute the high performance of SAPS mainly to the quite sophisticated design of our decoder, and in particular to its susceptibility to learning. This claim is supported by the comparison of SAPSpre VH Att (256) to SAPSpre VH Att (256) Fixed (Table 3), where the latter used a fixed decoder, pretrained in the autoassociative mode (Section 2.3). The huge gap between the performances of these architectures clearly indicates that it was essential to allow the decoder to adapt in the end-to-end learning spanning from NL to AST trees. The experiment with different signal propagation schemes (Table 2) suggests that the decoder owes its malleability in great part to the proposed 'liberal' state propagation techniques, in particular allowing the horizontal LSTM in DRNN to propagate its state both horizontally and vertically. Last but not least, the proposed soft attention mechanism also contributes significantly to the performance of SAPS, even though it has no access to the original NL specification, but only to the latent vector. This again points to high informativeness of the latent representation that emerged there during learning.

## 7 CONCLUSION

We have shown that reliable end-to-end synthesis of nontrivial programs from NL is plausible with exclusively neural means and gradient descent as the learning mechanism. Given that (i) capturing the semantics of the NL is in general difficult, (ii) there are deep differences between NL and AST trees on both syntactic and semantic level, and (iii) the correspondence between the parts of the former and the latter is far from trivial, we find this result a promising demonstration of the power dwelling in the neural paradigm. We find it likely that methods like SAPS may be successfully applied for practical synthesis of short code snippets, for instance in end-user programming. In future works, we plan to devise means to address the composite character of both specifications and program code.

**Acknowledgment**. We thank the authors of (Polosukhin & Skidanov, 2018) for publishing their code and data.

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
