# OpenReview forum: "Ain't Nobody Got Time for Coding: Structure-Aware Program Synthesis from Natural Language"
_ICLR.cc/2019/Conference_

### Official Review · AnonReviewer2 · 2018-10-30
**Autoencoder used for program synthesis**

**Rating:** 4
**Confidence:** 5

**Review:**

The submission proposes to combine a tree2tree autoencoder with a sequence encoder for natural language. It uses the autoencoding objective to appropriately shape the latent space and train the decoder, and then uses a second training step to align the output of a sequence encoder with the input for the tree decoder. Experiments on a recent dataset for the natural language-to-code task show that the proposed model is able to beat simple baselines.

There's much to like about this paper, but also many aspects that are confusing and make it hard to tease out the core contribution. I'm trying to reflect my understanding here, but the authors could improve their paper by providing an explicit contribution list. Overall, there seem to be three novel things presented in the paper:
(1) (Pre)training the (program) tree decoder using an autoencoder objective
(2) The doubly-recurrent tree decoder, which follows a different signal propagation strategy from most other approaches.
(3) An "attention" mechanism over the point in latent space (that essentially rescales parts of the decoder input)

However, the experiments do not evaluate these contributions separately; and so their relative merits remain unclear. Primarily, I have the following questions (for the rebuttal, and to improve the paper):

Re (1):
 (a) Does the pre-training procedure help? Did you evaluate joint end-to-end training of the NL spec encoder and the tree decoder?
 (b) The auto-encoder objective would allow you to train on a larger corpus of programs without natural language specifications. Arguably, the size of the dataset is insufficient for most high-capacity deep learning models, and as you use word embeddings trained on a much larger corpus...), you could imagine training the autoencoder on an additional corpus of programs without NL specs. Did you attempt this?

Re (2):
 (a) The tree decoder is unusual in that (one) part of the recurrence essentially enforces a breadth-first expansion order, whereas almost all other approaches use a depth-first technique (with the only exception of R3NN, as far as I remember). You cite the works of Yin & Neubig and Rabinovich et al.; did you evaluate how your decoder compares to their techniques? (or alternatively, you could compare to the absurdly complex graph approach of Brockschmidt et al. (arxiv 1805.08490)))
 (b) Ablations on this model would be nice: How does the model perform if you set the horizontal (resp. the vertical) input to 0 at each step? (i.e., ablations to standard tree decoder / to pure BFS)

Re (3): This is an unusual interpretation of the attention mechanism, and somewhat enforced by your choice (1). If you run an experiment on end-to-training (without the autoencoder objective), you could use a standard attention mechanism that attends over the memories of the NL encoder. I would be interested to see how this would change performance.

As the experimental evaluation seems to be insufficient for other researchers to judge the individual value of the paper's contribution, I feel that the paper is currently not in a state that should be accepted for publication at ICLR. However, I would be happy to raise my score if (some) of the questions above are answered; primarily, I just want to know if all of the contributions are equally important, or if some boost results more than others.


Minor notes:
- There are many spelling mistakes ("snipped" for "snippet", "isomorhpic", ...) -- running a spell checker and doing a calm read-through would help with these details.
- page1par2: Writing specifications for programs is never harder than writing the program -- a program is a specification, after all. What you mean is the hardness of writing a /correct/ and exact spec, which can be substantially harder. However, it remains unclear how natural language would improve things here. Verification engineers will laugh at you if you propose to "ease" their life by using of non-formal language...
- page1par3: This (and the rest of the paper) is completely ignoring the old and active field of semantic parsing. Extending the related work section to compare to some of these works, and maybe even the experiments, would be very helpful.
- page2par3 / page6par4 contradict each other. First, you claim that mostly normal english vocabulary is used, with only occasional programming-specific terms; later you state that "NL vocabulary used in specifications is strongly related to programming". The fact that there are only 281 (!!!) unique tokens makes it very doubtful that you gain anything from using the 1.9million element vocab of GLoVe instead of direct end-to-end training...
- page4par3: You state "a reference to a previously used variable may require 'climbing up' the tree and then descending" - something that your model, unlike e.g. the work of Yin & Neubig, does not support. How important is this really? Can you support your statement by data?
- page5, (14) (and (13), probably): To avoid infinite recursion, the $h_i^{(pred)}$ on the right-hand-side should probably be $h_{i-1}^{(pred)}$

---

> ### Author Response · Authors · 2018-11-26
> **Comment to AnonReviewer2**
>
> Dear Reviewer, please pardon our brevity, shorting your comments, and skipping minor comments - all due to 5000 char limit. Our responses start with '>' character. -- The authors.
>
> The authors could improve their paper by providing an explicit contribution list.
>
> > Thank you for your account of claimed contributions - we admit that the latter was too cursory in the original paper. The list you've provided above summarizes them very aptly, and the revised paper now features a very similar summary at the end of Introduction.
>
>
> Re (1):
>  (a) Does the pre-training procedure help? Did you evaluate joint end-to-end training of the NL spec encoder and the tree decoder?
>  (b) You could imagine training the autoencoder on an additional corpus of programs without NL specs. Did you attempt this?
>
> > Suggestion (1a) has been voiced also by another reviewer, and we conducted now an additional experiment to verify this hypothesis. Suggestion (1b) is very interesting, and was indeed one of conceptual arguments in favor of engaging autoassociative learning. We will definintely try ot give it a go in future, however at the moment we don't have an auxiliary dataset that could be used for that purpose (or an appropriate program generator). Also, we present a range of other results in the revised version, and squeezing yet another experiment would be problematic due to page limit. Nevertheless, that's a very interesint hint for follow-ups, thank you.
>
>
> Re (2):
>  (a) The tree decoder is unusual in that (one) part of the recurrence essentially enforces a breadth-first expansion order, whereas almost all other approaches use a depth-first technique (with the only exception of R3NN, as far as I remember). You cite the works of Yin & Neubig and Rabinovich et al.; did you evaluate how your decoder compares to their techniques? (or alternatively, you could compare to the absurdly complex graph approach of Brockschmidt et al. (arxiv 1805.08490)))
>  (b) Ablations on this model would be nice: How does the model perform if you set the horizontal (resp. the vertical) input to 0 at each step? (i.e., ablations to standard tree decoder / to pure BFS)
>
> > Thank you for this suggestion. Lack of ablation studies was a recurring theme in virtually all reviewes we've received, so we conducted a few of them and report them now in the revised version. Enabling and disabling the state propagation mechanism separately for vertical and horizontal propagation is now one of the main dimensions of the experimental part of the paper.
>
>
> If you run an experiment on end-to-training (without the autoencoder objective), you could use a standard attention mechanism that attends over the memories of the NL encoder.
>
> > This is indeed possible, once the decoder has been extracted from the autoencoder. Nevertheless, we decided not to experiment with this specific extension, as page limit was already problematic with the other new results we now include.
>
>
> This (and the rest of the paper) is completely ignoring the old and active field of semantic parsing.
>
> > Thank you for this very valuable pointer. We now cite a related work on semantic parsing.
>
>
> - page2par3 / page6par4 contradict each other.
>
> > What we meant in page2par3 is that the dataset includes hardly any out-of-vocabulary terms. But we did not phrase that correctly indeed and the contradiction is rather evident - now fixed.
>
>
> - page4par3: You state "a reference to a previously used variable may require 'climbing up' the tree and then descending" - something that your model, unlike e.g. the work of Yin & Neubig, does not support. How important is this really? Can you support your statement by data?
>
> > Admittedly not. This was indeed largely speculative. However, we did not mean to suggest that we can explicitly climb the tree, but that non-resetting of network's state provides the decoder with more context. We replaced that sentence with a more accurate statement.
>
>
> - page5, (14) (and (13), probably): To avoid infinite recursion, the $h_i^{(pred)}$ on the right-hand-side should probably be $h_{i-1}^{(pred)}$
>
> > Formula 14 is used to modify h_i^{(pred)}, computed earlier in Formula 8. It indeed could be advantegous to emphasise it, however in the case of SAPS $h_{i-1}^{(pred)}$ would denote the hidden state of the previous node rather than the hidden state belonging to i-th node, but from previous timestep. Due to that, we decided to leave this formula unchanged.
>
> > Thank you for these very precise ad insightful remarks. We took them all into account.

---

> > ### Comment · AnonReviewer2 · 2018-11-27
> > **Updated comments after revision**
> >
> > Thank you for the changes to the paper. I think they have already greatly improved the quality of your submission. First, let me directly answer two points that you've made:
> >
> > Re (1): The results in the new revision show that pre-training the decoder in the autoencoder setting has no noticeable effect on the effectiveness of the proposed model, meaning that there is no empirical evidence that the autoencoder has practical value. Without experiments on additional data used for this pretraining (1(b) in my original review), it thus remains unclear why this modelling choice was made.
> >
> > Re (2): The results in Table 2 are extremely surprising: Tree decoders with top-down/vertical propagation schemes (e.g., Rabinovich et al. 2017, Chen at al. 2018) are known to have results that are comparable or better than sequence-based baselines in the generation of programs. It is unclear why these do not work at all here, and the authors do not explain this disagreement with the literature.
> >
> > Minor note: Encoding a NL spec to a latent representation and then decoding a program from that is not an original contribution; it is a well-known, central idea in many deep learning-based semantic parsing works of the last few years.
> >
> > The combination of these issues is reason enough for me keep my rating at 4: Of the contributions claimed by the authors (in the revised version), (i) is not original, (ii) is shown by their experiments to have no effect, (iii) seems to be not evaluated properly (and not compared to any non-trivial existing baselines in the space). This leaves contribution (iv) (attention over dimensions of latent spec), which seems to have a small effect in the experiments. For me, this is not enough to recommend acceptance of the paper, especially one that is dominated by other content that is less novel or interesting.

---

### Official Review · AnonReviewer1 · 2018-11-02
**Reasonable model but unclear results**

**Rating:** 4
**Confidence:** 3

**Review:**

# Summary

This paper introduces a model called SAPS for the task of mapping natural language descriptions of programs to the AST tree of the corresponding program. The model consists of a variation of a double recurrent neural network (DRNN) which is pre-trained using an autoencoder. The natural language description is turned into a latent vector using pretrained word embeddings and a bidirectional stacked LSTM. The final model consists of training this sentence embedding model jointly with the decoder of the autoencoder.

# Quality

The authors introduce a reasonable model which achieves good performance on a relevant task. The results section contains a fair amount of numbers which give some insight into the performance of the model. However, the results make it hard to compare the proposed model with other models, or with other training procedures.

For example, the Seq2Tree model that is shown in table 2 was not necessarily intended to be used without a search algorithm. It is also not mentioned how many parameters both models have which makes it hard to judge how fair of a comparison it is. (I couldn't find the dimensionality of the encoder in the text, and the decoder dimensionality is only shown in figure 2.)

The model proposed in this work uses decoder pretrained in an autoencoder setting. No results are shown for how the model performs without pretraining. Pretraining using autoencoders is a technique that fell out of favor years ago, so it seems worthwhile to investigate whether or not this pretraining is necessary, and if so, why and how it aids the final performance.

It is unclear to me what type of robustness the authors are trying to show in table 5. The use of robustness here is not clear (robustness is often used to refer to a network's susceptability to adversarial attacks or perturbations of the weights). It also seems that the type of "simple replacements" mentioned are very similar to the way the examples were generated in the first place (section 4 of Polosukhin). If the goal is to measure generalization, why do the authors believe that performance on the test set alone is not a sufficient measure of generalization?

Some smaller comments and questions:

* In section 4 you mention training the sentence-to-tree and the sentence-to-vector mappings. Isn't the sentence-to-vector model a subset of the sentence-to-tree model? Should I interpret this as saying that, given the pretrained decoder and the glove embeddings, you now train the entire model jointly? Or do you first learn the mapping from sentences to the latent space, and only then finetune the entire model?
* The attention mechanism is not actually an attention mechanism: Attention mechanisms are used to reduce a variable number of elements to a single element by learning a weighting function and taking a weighted sum. My understanding is that in this case, the input (the latent representation) is of fixed size. The term "gating function" would be more appropriate.
* You specify that the hidden states of the decoder are initialized to zero, but don't specify what the cell states are initialized to.

# Clarity

The writing in the paper is passable. It lacks a bit in structure (e.g., I would introduce the problem and dataset before introducing the model) and sometimes fails to explain what insights the authors draw from certain results, or why certain results are reported. Take table 3 as an example: As a reader, I was confused at first why I should care about the reconstruction performance of the autoencoder alone, considering its only purpose is pretraining. Then, when looking at the numbers, I am even more confused since it is counterintuitive that it is harder to copy a program than it is to infer it. At the end of the paragraph the authors propose an explanation (the encoder isn't as powerful as the decoder) but leave it unclear as to why these numbers were being reported in the first place.

In general, the paper would do well to restructure the text so that the reader is explained what the goal of the different experiments is, and what insights should be drawn from them.

A variety of smaller concerns and comments:

* Please reduce and harmonize the terminology in the paper: the terms latent-to-AST, NLP2Tree, NLP2Vec, tree2tree/tree-to-tree, sentence-to-tree, sentence-to-vector, NL-to-latent, and spec-to-latent all appear in the paper and several of them are redundant, making it significantly harder to follow along with the text.
* Avoid citing the same work multiple times within a paragraph; cite only the first use and use prose to make clear that future references are to the same work.
* Formula 14 has h_i^{(pred)} on both sides of the quation, and is used in the definition of A as well. I am assuming these two terms are actually the h_i^{(pred)} from equation 8, but this should be made clear in the notation.
* Why does figure 1 have boxes for "NL specification", "NL spec.", and "NL query"? In general, the boxes inconsistently seem to represent both values and operations.
* It is never explicitly stated that Seq2Tree is the model from the Polosukhin et al. paper, which is a bit confusing.
* Parameters is missing an -s in the first paragraph of section 4.
* It is said that regularization is applied to layer normalization, which I assume means that the regularization is applied to the gain parameters of the layer normalization.
* It says "like a in the above example" when table 1 is rendered at the bottom of the page by Latex.

# Originality and significance

The paper introduces model variations that the authors claim improve performance on this particular program synthesis problem. In particular, in the DRNN decoder the hidden state is never reset for the "horizontal" (breadth-first order) decoder, and each node is only allowed to attend over the latent representation of the program. The authors claim this means their model "significantly diverges from [other] works", which seems hyperbolical.

The main contribution of this work is then the performance on the program synthesis task of Polosukhin and Skidanov. However, the model fails to improve on the Seq2Tree-guided search approach, so its main claimed benefit is that it is trained end-to-end. Although there is a strong trend in ML research to prefer end-to-end systems, it is worthwhile to ask when and why end-to-end systems are preferred. It is often clear that they are better than having separately learned components that are combined later. However, this does not apply to the model from Polosukhin et al., which consists of a single learned model being used in a search, which is a perfectly acceptable technique. In comparison, translations in neural machine translation are also produced by performing a beam search, guided by the probabilities under the model.

The paper would be significantly stronger if it could show that some alternative/existing method (e.g., a standard DRNN, or a Seq2Tree model with the same number of parameters, or a non-pretrained network) would fail to solve the problem where the authors' proposed method does not. However, the single comparison with the Seq2Tree model does not show this.

# Summary

Pros:

* Reasonable model, extensive results reported
* Decently written

Cons:

* Unclear how the performance compares to other models
* Not well justified why end-to-end methods would be better than guided-search based methods
* Model architectural differences seem relatively minor compared to original DRNN
* The pretraining using an autoencoder and the use of pretrained word embeddings seems arbitrary and is not critically evaluated
* Lack of coherent story to several results (the autoencoder performance, robustness analysis)

---

> ### Author Response · Authors · 2018-11-26
> **Comment to AnonReviewer1**
>
> Dear Reviewer, please pardon our brevity, shorting your comments, and skipping minor comments - all due to the 5000 char limit. Our responses start with '>' character.  -- The authors.
>
> The results make it hard to compare the proposed model with other models, or with other training procedures.
>
> > In the original contribution, we made our best effort to make the approach as comparable as possible to Polosukhin & Skidanov (2018). In the attached revision, we sustain this perspective, but enrich the paper with additional results that will hopefully meet your expectations.
>
> For example, the Seq2Tree model that is shown in table 2 was not necessarily intended to be used without a search algorithm.
>
> > Agreed, though we don't think we ever claimed so in the original paper. Arguably, any seq2seq and tree2tree model can be used both with or without a search algorithm, and we think that testing them in both these settings is valuable. Polosukhin & Skidanov must have been of the same opinion here, given that they reported results in both settings.
>
> It is also not mentioned how many parameters both models have which makes it hard to judge how fair of a comparison it is.
>
> > Indeed. We include now the sizes of layers rather.  The total number of parameters is 5.73 mln (SAPS256 + HV + Att).
>
> No results are shown for how the model performs without pretraining.
>
> > We conducted accordingly additional experiments and report them now in the revised version in Section 4.1.
>
> Or do you first learn the mapping from sentences to the latent space, and only then finetune the entire model?
>
> > In the original submission, we did exactly as you suggest above: trained the sentence-tree mapping end-to-end, with fixed Glove embedding and starting from the decoder taken from the autoencoder. In the revised version, we lay out these variants more clearly: there are models trained from scratch, pretrained, and those that use pretrained decoder that remains fixed in further training.
>
> The attention mechanism is not actually an attention mechanism.
>
> > Thank you, we added now `gating function' as an alternative term to ease understanding. However, we'd still claim that this _is_ a form of attention, though in a _soft_ meaning of that word - as it is usually meant for instance in attention mechanisms used for image analysis in convolutional NNs (soft vs hard attention).
>
> The writing in the paper is passable. It lacks a bit in structure.
>
> > We revised the text thoroughly. Concerning your specific remark on Table 3 (now Table 4), we rephrased the corresponding paragraph so that it is now trying to anticipate reader's expectations.
>
> Please reduce and harmonize the terminology in the paper.
>
> > We made the terminology more consistent; most of the terms used in the paper reflect now the building blocks shown in Fig. 1.
>
> Formula 14 has h_i^{(pred)} on both sides of the equation, and is used in the definition of A as well.
>
> > Formula 14 can be seen as a development of Formula 8: We first compute the 'raw' version of h_i^{(pred)} in Formula 8, then slightly modify it using both h^{latent} and previously computed h_i^{(pred)}.
>
> Why does figure 1 have boxes for "NL specification", "NL spec.", and "NL query"?
>
> > Fig. 1 has been redrawn and the names of building blocks are now consistent with the text.
>
> It is said that regularization is applied to layer normalization, which I assume means that the regularization is applied to the gain parameters of the layer normalization.
>
> > Indeed there was an inconsistency: we meant that the regularization is applied to all parameters excluding biases and parameters corresponding to layer normalization.
>
> The authors claim this means their model "significantly diverges from [other] works", which seems hyperbolical.
>
> > Please notice that that sentence originally read: "However it significantly diverges from those works in using only the latent vector for passing the information between encoder and decoder." As you can see, our emphasis was on the fact that that process uses only the latent layer as its source of information. Nevertheless, we removed the adverb `significantly' from that sentence.
>
> However, the model fails to improve on the Seq2Tree-guided search approach, so its main claimed benefit is that it is trained end-to-end.
>
> > Competing with search-based methods was not our main point. We find it rather obvious that explicit search is a powerful mechanism and should in principle always lead to improvements when used in combination with some base method. Our argument for focusing on neural end-to-end learning was primarily of philosophical nature: it seems very compelling to how far can we go by expressing a program as a point in a fixed-dimensional latent space. Some other arguments (possibility of training autoencoder on unlabeled examples, elegance, etc) are now discussed in Discussion.

---

> > ### Comment · Area_Chair1 · 2018-11-30
> > **Paging R1**
> >
> > Could R1 kindly please take a look at the rebuttal and changes to the paper the authors have made and consider whether they would like to revise their assessment of the paper, or if sticking to their position, give a quick explanation as to why their score would remain unchanged.

---

> > > ### Comment · AnonReviewer1 · 2018-12-02
> > > **Not updating score**
> > >
> > > Thanks to the authors for their rebuttal and changes to the paper. The changes address most of the issues I had with clarity and reporting of experimental results.
> > >
> > > However, I don't think these changes address the main shortcoming, which is that the paper fails to make a clear contribution. The architectural modifications and training procedure (autoencoder pretraining) are not very strongly justified given the experimental results. On the other hand, the observation that a fixed size representation can work quite well does not seem very surprising (the same is true for models in, e.g., neural machine translation). And the final results on a relatively synthetic dataset don't suggest that this model achieves any large improvements compared to existing models.

---

### Official Review · AnonReviewer3 · 2018-11-03
**Interesting ideas, but no ablation experiments to attribute usefulness of the ideas**

**Rating:** 4
**Confidence:** 4

**Review:**

This paper proposes the Structure-Aware Program Synthesis (SAPS) system, which is an end-to-end neural approach to generate snippets of executable code from the corresponding natural language descriptions. Compared to a previous approach that used search in combination of a neural encoder-decoder architecture, SAPS relies exclusively on neural components. The architecture uses a pretrained GloVe embedding to embed tokens in the natural language description, which are then embedded into a vector representation using a bidirectional-LSTM. The decoder uses a doubly-recurrent neural network for generating tree structured output. One of the key ideas of the approach is to use a single vector point in the latent space to represent the program tree, where it uses a tree2tree autoencoder to pre-train the tree decoder. The results on the NAPS dataset show an impressive increase in accuracy of about 20% compared to neural-only baselines of the previous approach.

While overall the SAPS architecture achieves impressive practical results, some of the key contributions to the design of the architecture are not evaluated and therefore it makes it difficult to attribute the usefulness and impact of the key contributions. For example, what happens if one were to use pre-trained GloVe embeddings for embedding NL specifications in Polosukhin & Skidanov (2018). Such and experiment would lead to a better understanding of how much gain in accuracy one can obtain just by using pre-trained embeddings.

One of the key ideas of the approach is to use a tree2tree autoencoder to train the latent space of program trees. The decoder weights are then initialized with the learnt weights and fine-tuned during the end-to-end training. What happens if one keeps the decoder weights fixed while training the architecture from NL descriptions to target ASTs? Alternatively, if one were to not perform auto-encoding based decoder pre-training and learn the decoder weights from scratch, how would the results look?

Another key point of the paper is to use a soft attention mechanism based on only the h^latent. What happens if the attention is also perform on the NL description embeddings? Presumably, it might be difficult to encode all of the information in a single h^latent vector and the decoder might benefit from attending over the NL tokens.

It was also not clear if it might be possible to perform some form of a beam search over the decoded trees to possibly improve the results even more?

There are also other datasets such as WikiSQL and Spider for learning programs from natural language descriptions. It might be interesting to evaluate the SAPS architecture on those datasets as well to showcase the generality of the architecture.

---

> ### Author Response · Authors · 2018-11-26
> **Comment to AnonReviewer3**
>
> Dear Reviewer, our responses start with '>' character. -- The authors.
>
>
> While overall the SAPS architecture achieves impressive practical results, some of the key contributions to the design of the architecture are not evaluated and therefore it makes it difficult to attribute the usefulness and impact of the key contributions. For example, what happens if one were to use pre-trained GloVe embeddings for embedding NL specifications in Polosukhin & Skidanov (2018). Such and experiment would lead to a better understanding of how much gain in accuracy one can obtain just by using pre-trained embeddings.
>
> > We admit that the experiments reported in the original submission provided limited insight into the inner workings of the method, in particular about the impact of individual components. In the revised paper, we examine more configurations (primarily ablations of the complete architecture) and analyze them from multiple angles. Unfortunately, the particular experiment you mention above could not be conducted, as we did not reimplement Polosukhin & Skidanov's method - we only quote the results from their paper (that was not only easier, but also arguably the only way to go, given (i) the complexity of the reference method, (ii) the fact that it's description of the paper was not precise enough at places, and (iii) the code provided by Polosukhin & Skidanov (2018) is complex and written in a framework we do not have experience in. Therefore it could be troublesome for us to modify their code and make sure that the crucial parts remain unchanged. Nevertheless, we hope that the presence other results we conducted compensates that shortcoming.
>
> One of the key ideas of the approach is to use a tree2tree autoencoder to train the latent space of program trees. The decoder weights are then initialized with the learnt weights and fine-tuned during the end-to-end training. What happens if one keeps the decoder weights fixed while training the architecture from NL descriptions to target ASTs? Alternatively, if one were to not perform auto-encoding based decoder pre-training and learn the decoder weights from scratch, how would the results look?
>
> > Thanks for these interesting proposals. We conducted additional experiments that address these questions and report their results in Section 4.
>
>
> Another key point of the paper is to use a soft attention mechanism based on only the h^latent. What happens if the attention is also perform on the NL description embeddings? Presumably, it might be difficult to encode all of the information in a single h^latent vector and the decoder might benefit from attending over the NL tokens.
>
> > That's indeed an interesting point, raised also by the other reviewers. Let us just emphasize here that strict separation of the encoding and decoding was an important conceptual assumption behind our approach. In particular, it allowed us to take the trained AST decoder as-is and combine it with an NL encoder. Nevertheless, you're right that there are no technical obstacles for linking the AST decoder to an attention mechanism on of the NL encoder. However, as we conducted also a range of other new experiments, squeezing in yet another one would be problematic due to the page limit. We would like to consider this extension in follow-up studies.
>
>
> It was also not clear if it might be possible to perform some form of a beam search over the decoded trees to possibly improve the results even more?
>
> > We are quite positive, if not certain, that such an extension would bring further improvements, per analogy to the experience of Polosukhin & Skidanov. However, the whole point of our attempt was to find out if effective synthesis from NL is possible with bare neural means. Therefore, we decided to maintain this perspective in the revised version. Even if we decided to consider and present also some beam search result, that would be technically challenging due to the required page limit. The revised paper should convey now this viewpoint, hopefully clear enough.
>
>
> There are also other datasets such as WikiSQL and Spider for learning programs from natural language descriptions. It might be interesting to evaluate the SAPS architecture on those datasets as well to showcase the generality of the architecture.
>
> > Thank you for these interesting pointers, not all of which we were aware of. We would be very happy to do so in a further perspective (like follow-up journal paper), but, again, page limit seems to be precluding that. Also, we intentionally focused here on one-to-one comparison to Polosukhin & Skidanov (2018), as we find it most state-of-the-art, and actually the only study that used that valuable dataset at the time we had started this project.

---

> > ### Comment · Area_Chair1 · 2018-11-30
> > **Paging R3**
> >
> > Could R3 kindly please take a look at the rebuttal and changes to the paper the authors have made and consider whether they would like to revise their assessment of the paper, or if sticking to their position, give a quick explanation as to why their score would remain unchanged.

---

> > > ### Comment · AnonReviewer3 · 2018-12-04
> > > **Thanks for the changes, but the key contributions are still not clear.**
> > >
> > > Thanks for the revision and the additional ablation experiments, which help quite a bit.
> > >
> > > The key contributions for the paper are still not that clear to me. One of the main contributions of using the auto-encoder for pre-training the decoder does not seem to help much (resulting in a slightly lower accuracy on the test set). I also think evaluating the Polosukhin & Skidanov (2018) model with pre-trained GloVe embeddings for embedding NL specifications would be important as well to make cleaner comparisons with the previous approaches. My score, therefore, unfortunately remains unchanged.

---

### Author Response · Authors · 2018-11-26
**To all reviewers**

Dear Reviewers,

We'd like to thank you for thorough analysis of our submission and many insightful observations. We found them very useful for improving the paper. We made our best effort to address your requests, and hope that the revised version will be considered sufficiently valuable to be accepted for ICLR.

Most importantly, in an attempt to address your most fundamental doubts, we conducted a range of new experiments and discuss them in the revised submission. Even though we do no not portray them as ablations in the paper, they may be seen as such w.r.t. to the most sophisticated SAPS configuration (pretraining, attention, and both directions of state propagation in decoder), so we hope they address the issues you pointed to in your reviews.

In connection to that, we revised the text, which at places required quite essential changes, in particular:
- shortening of Introduction,
- swapping of sections 2.2 and 2.3,
- reordering the presentation of experimental results and rewriting most of the comments to those results (note a new table and a different ordering of tables),
- revising some threads in Discussion.

You'll find our detailed responses in individual files. Please note that your comments are quoted as-is, while our responses start with the `>' mark. Excuse our brevity there - it was sometimes hard to meet the 5000 char limit, so in some cases we had to shorten or select your comments too.

Best regards,
The authors

---

### Meta-Review · Area_Chair1 · 2018-12-13
**More experimental rigour is needed**

**Confidence:** 5
**Recommendation:** Reject

**Metareview:**

The paper presents a new neural program synthesis architecture, SAPS, which seems to produce accuracy improvements in some synthesis tasks. The reviewer consensus, even after discussion with the authors, was that the paper is not acceptable at the conference. Two concerns emerge during discussion, even considering the authors efforts to improve the paper. First, the system seems to have many "moving parts", but there is a lack of rigorous ablation studies to demonstrate which components of the system (or combination thereof) make significant contributions to the results. I agree with this assessment: it is not sufficient to demonstrate increased scores, even if the experimental protocol and clear and sound (more on this later), but there must be some evidence as to why this increase happens, both in the discussion and in the empirical segment of the paper, by conducting a thorough ablation study. Second, all reviewers had issues with proper and fair comparison with prior work, with the consensus being that the model is not adequately compared to convincing benchmarks in the paper.

The results of the paper sound like there is something promising going on, but the need for a clear presentation of what is the driving factor behind any improvement is not only a superficial stylistic requirement, but a key tenet of proper scholarship. This is one front on which the paper fails to make a successful case for the work and methods it describes, and unfortunately is not ready for publication at this time (despite having a cool title).